# Cannabidiol Modulates M-Type K^+^ and Hyperpolarization-Activated Cation Currents

**DOI:** 10.3390/biomedicines11102651

**Published:** 2023-09-27

**Authors:** Yen-Chin Liu, Edmund Cheung So, Sheng-Nan Wu

**Affiliations:** 1Department of Anesthesiology, Kaohsiung Medical University Hospital, Kaohsiung 80756, Taiwan; anesliu@kmu.edu.tw; 2Department of Anesthesiology, School of Post-Baccalaureate, College of Medicine, Kaohsiung Medical University, Kaohsiung 80756, Taiwan; 3Department of Anesthesiology, National Cheng Kung University Hospital, College of Medicine, National Cheng Kung University, Tainan 701401, Taiwan; 4Department of Anesthesia, An-Nan Hospital, China Medical University, Tainan 70965, Taiwan; 5Department of Physiology, National Cheng Kung University Medical College, Tainan 70101, Taiwan; 6School of Medicine, National Sun-Yat Sen University College of Medicine, Kaohsiung 80424, Taiwan; 7Department of Research and Education, An-Nan Hospital, China Medical University, Tainan 70965, Taiwan

**Keywords:** cannabidiol (CBD), M-type K^+^ current, hyperpolarization-activated cation current, *erg*-mediated K^+^ current, voltage-gated Na^+^ current, pituitary cell

## Abstract

Cannabidiol (CBD) is a naturally occurring compound found in the *Cannabis* plant that is known for its potential therapeutic effects. However, its impact on membrane ionic currents remains a topic of debate. This study aimed to investigate how CBD modifies various types of ionic currents in pituitary GH_3_ cells. Results showed that exposure to CBD led to a concentration-dependent decrease in M-type K^+^ currents (*I*_K(M)_), with an IC_50_ of 3.6 μM, and caused the quasi-steady-state activation curve of the current to shift to a more depolarized potential with no changes in the curve’s steepness. The CBD-mediated block of *I*_K(M)_ was not reversed by naloxone, suggesting that it was not mediated by opioid receptors. The *I*_K(M)_ elicited by pulse-train stimulation was also decreased upon exposure to CBD. The magnitude of *erg*-mediated K^+^ currents was slightly reduced by adding CBD (10 μM), while the density of voltage-gated Na^+^ currents elicited by a short depolarizing pulse was not affected by it. Additionally, CBD decreased the magnitude of hyperpolarization-activated cation currents (*I*_h_) with an IC_50_ of 3.3 μM, and the decrease was reversed by oxaliplatin. The quasi-steady-state activation curve of *I*_h_ was shifted in the leftward direction with no changes in the slope factor of the curve. CBD also diminished the strength of voltage-dependent hysteresis on *I*_h_ elicited by upright isosceles-triangular ramp voltage. Collectively, these findings suggest that CBD’s modification of ionic currents presented herein is independent of cannabinoid or opioid receptors and may exert a significant impact on the functional activities of excitable cells occurring in vitro or in vivo.

## 1. Introduction

Cannabidiol (CBD) is a non-psychoactive cannabinoid derived from the *Cannabis* plant, known for its potential therapeutic effects. It is among over 100 cannabinoids present in the plant and has been shown to be effective in treating various medical conditions, such as epilepsy, bipolar disorder, inflammation, and cancer [1,2,3,4,5]. Recent studies have demonstrated that CBD can modify the activity in the hypothalamic–pituitary–adrenal axis [6,7] and can modulate different types of transmembrane ionic currents in electrically excitable cells, including the voltage-gated Na^+^ current (*I*_Na_) and the M-type K^+^ currents (*I*_K(M)_) [5,8,9]. Thus, further clarification is necessary to determine the effects of CBD or related compounds on ionic currents present in the membranes of electrically excitable cells.

The KCNQ2, KCNQ3, and KCNQ5 genes encode the core subunits of K_V_7.2, K_V_7.3, and K_V_7.5 channels, respectively [10]. These potassium (K^+^) channels, when activated, generate the M-type K^+^ current (*I*_K(M)_), which is characterized by its low threshold voltage activation and by slow activation and deactivation properties [11,12]. The modulation of *I*_K(M)_ has gained significant recognition as an additional therapeutic approach for treating a range of neurological disorders characterized by excessive neuronal activity. These disorders encompass conditions like cognitive dysfunction, neuropathic pain, and epilepsy [10,13,14,15]. Furthermore, the magnitude of *I*_K(M)_ is believed to regulate the availability of voltage-gated Na^+^ (Na_V_) channels during prolonged high-frequency firing [16]. Previous studies have shown that CBD can modify the magnitude of *I*_K(M)_ [9,17]. However, it is still largely unclear whether and how exposure to CBD can perturb the magnitude or gating properties of *I*_K(M)_.

The hyperpolarization-activated cation current (*I*_h_), also known as the “funny current” (*I*_f_), plays a crucial role in regulating repetitive electrical activity in cardiac cells, various types of central neurons, and endocrine or neuroendocrine cells [11,18,19,20,21]. This ionic current exhibits unique characteristics, including slow voltage-dependent activation kinetics and a mixed Na^+^/K^+^ current that flows inwardly, and it can be blocked by CsCl or ivabradine [20,22]. Activation of *I*_h_ may lead to depolarization of the resting potential, reaching the threshold required for generating or triggering an action potential. Consequently, it influences pacemaker activity and impulse propagation [22]. Additionally, the slow kinetics of *I*_h_ in response to prolonged hyperpolarization can result in long-lasting, activity-dependent modulation of membrane excitability across various types of excitable cells [23]. *I*_h_ is mediated by channels encoded by members of the hyperpolarization-activated cyclic-nucleotide-gated (HCN) gene family, and studies have demonstrated that the activity of these channels underlies the ionic mechanisms associated with both convulsive disorders and inflammatory pain disorders [24,25,26].

Therefore, based on the aforementioned information, our aim was to investigate the effects of CBD or other related compounds on perturbations in various ionic currents (such as *I*_Na_, *I*_K(M)_, *I*_h_, and *I*_K(erg)_) in pituitary GH_3_ cells. The findings from this study highlight the evidence showing that CBD can directly and effectively inhibit the magnitude of *I*_K(M)_ and *I*_h_ in a concentration- and voltage-dependent manner. Such suppression of ionic currents appears to be direct and is highly unlikely to be linked to its binding to cannabinoid receptors.

## 2. Materials and Methods

### 2.1. Chemicals, Drugs, and Solutions Used for This Work

Cannabidiol (CBD, 2-[(1R,6R)-3-methyl-6-(prop-1-en-2-yl)cyclohex-2-en-1-yl]-5-pentylbenzene-1,3-diol) and cannabichromene were acquired from Sigma-Aldrich (Cerilliant Corp., Darmstadt, Germany). Linopirdine (Lino), naloxone, NS1643, oxaliplatin (OXAL), tefluthrin (Tef), tetraethylammonium chloride (TEA), tetrodotoxin (TTX), and thyrotropin-releasing hormone (TRH) were supplied by Sigma-Aldrich (Merck, Darmstadt, Germany). Liraglutinide was obtained from Novo Nordisk^®^ (Bagsværd, Denmark), telmisartan (TEL) was obtained from Tocris (Bristol, UK), and liraglutinide (Saxenda^®^) was obtained from MedChemExpress (Genechain, Kaohsiung, Taiwan).

Unless stated otherwise, the cell culture materials, such as Ham’s F-12 medium, L-glutamine, horse serum, and fetal calf serum, were obtained from HyClone^TM^ (Thermo Fisher; Level Biotech, Tainan, Taiwan). All other chemicals and reagents, such as CdCl_2_, CsCl, CsOH, HEPES, and aspartic acid, were commercially available and of analytical reagent grade.

The HEPES-buffered normal Tyrode’s solution used in this study had the following composition (in mM): NaCl 136.5, KCl 5.4, CaCl_2_ 1.8, MgCl_2_ 0.53, glucose 5.5, and HEPES-NaOH buffer 5.5 (pH 7.4). For recording *I*_h_, *I*_K(M)_, or *I*_K(erg)_, the recording pipettes were filled up with the following solution (in mM): K-aspartate 130, KCl 20, KH_2_PO_4_ 1, MgCl_2_ 1, EGTA 0.1, Na_2_ATP 3, Na_2_GTP 0.1, and HEPES-KOH buffer 5 (pH 7.2). To measure whole-cell *I*_K(M)_ or *I*_K(erg)_, high-K^+^ bathing solution was used with the following composition (in mM): KCl 145, MgCl_2_ 0.53, and HEPES-KOH 5 (pH 7.4). All solutions were prepared using deionized water obtained from a Milli-Q^®^ water purification system (APS Water Services, Inc., Van Nuys, CA, USA). On the day of use, the pipette solution and culture media were filtered through an Acrodisc^®^ syringe filter with 0.2 μm of Supor^®^ membrane (Palll Bio-Check, Tainan, Taiwan).

### 2.2. Preparation of Pituitary GH_3_ Cells

GH_3_ pituitary tumor cells, originally derived from ATCC [CCL-82^TM^] and obtained from the Bioresources Collection and Research Center (BCRC-60015) in Hsinchu, Taiwan, were cultured in Ham’s F-12 medium supplemented with 15% (*v*/*v*) horse serum, 2.5% (*v*/*v*) fetal calf serum, and 2 mM L-glutamine in a humidified environment with 5% CO_2_/95% air [19]. The verification of GH_3_ cells was conducted by measuring the level of prolactin in the culture medium. Experiments were conducted 5 to 6 days after the cells had reached 60–80% confluence.

### 2.3. Electrophysiological Recordings Using Patch-Clamp Technique

The GH_3_ cells were dispersed just before each measurement with care. Then, a small amount of the cell suspension was rapidly transferred to a custom-made chamber, where the cells were allowed to settle on the bottom surface. A DM-IL inverted microscope (Leica; Highrise Instrument, Taichung, Taiwan) was used to monitor cell size during the experiments. A custom-made chamber was tightly placed on the microscope’s stage, and a video camera system with a magnification of 1500× was connected to the microscope. The cells that were examined were maintained in a bath of normal Tyrode’s solution with a concentration of 1.8 mM CaCl_2_, at room temperature (20–25 °C). 

The patch electrodes were meticulously fashioned from Kimax-51 capillaries with an outer diameter of 1.5–1.8 mm (#34500; Kimble, Dogger, New Taipei City, Taiwan) using a two-stage PP-830 puller (Narishige; Taiwan Instrument, Tainan, Taiwan). The electrode tips were then fire-polished using an MF-83 microforge (Narishige). Typically, the filled electrodes had tip resistances ranging from 2 to 4 MΩ. For standard patch-clamp recordings in a modified whole-cell configuration [19,20], we employed an RK-400 patch amplifier (Bio-Logic, Claix, France). The cell capacitance was measured to be 34 ± 6 pA (*n* = 32). Junction potentials, which arise at the electrode tip due to differences in composition between the internal solution and the bath solution, were nullified shortly before establishing a seal formation. Junction potential corrections were subsequently applied to the whole-cell data. During the recording process, the signal output data (i.e., potential or current tracings) were acquired and stored online at a frequency of 10 kHz or higher kHz. This was achieved using an ASUSPRO-BN401 LG laptop computer (ASUS, Yuan-Dai, Tainan, Taiwan) equipped with a Digidata 1440A (Molecular Devices; Advanced Biotech, New Taipei City, Taiwan). The entire system was controlled by pClamp 10.6 software (Molecular Devices).

### 2.4. Analyses of Whole-Cell Recordings

We analyzed the concentration–response curves of CBD-induced inhibition on the density of *I*_K(M)_ or *I*_h_ in GH_3_ cells. To elicit the slowly activating *I*_K(M)_, cells were placed in high-K^+^, Ca^2+^-free solution, and we applied a 1 s depolarizing pulse to −10 mV from a holding potential of −50 mV, as described previously [11]. The current densities were measured at the end of the depolarizing pulse in the presence or absence of exposure to different CBD concentrations. To evoke *I*_h_, cells were bathed in Ca^2+^-free Tyrode’s solution and subjected to a 2 s hyperpolarizing pulse to −110 mV from a holding potential of −40 mV. The *I*_h_ density was measured with or without CBD at the end of the hyperpolarizing pulse. The concentration of CBD needed to inhibit 50% of the *I*_K(M)_ or *I*_h_ (i.e., IC_50_) was approximated using a Hill function as follows: percentage  decrease%=[CBD]nH×EmaxIC50nH+[CBD]nH
where [CBD] is the CBD concentration applied, IC_50_ is the half-maximal concentration of CBD, n_H_ is the Hill coefficient, and E_max_ is the maximal decrease in *I*_K(M)_ or *I*_h_ caused by the CBD presence.

To analyze the steady activation curve of *I*_K(M)_ or *I*_h_ in the absence and presence of CBD, we fitted the data using a Boltzmann function. The Boltzmann equation is defined as: GGmax=11+exp±V−V1/2/k
where G_max_ is the maximal conductance of either *I*_K(M)_ or *I*_h_ acquired from the absence or presence of 3 μM CBD, and *V* and *V*_1/2_ represent the membrane potential in mV and the half-point of the activation curve of *I*_K(M)_ or *I*_h_, respectively. The slope factor of the activation of *I*_K(M)_ or *I*_h_ is represented by *k*. 

### 2.5. Methods for Curve-Fitting and Statistical Analyses

Continuous curves were fitted to the experimental data using linear or nonlinear regression methods, such as exponential or sigmoidal function. The software tools used for this purpose included pClamp 10.6 software (Molecular Devices), 64-bit OriginPro 2022b software (OriginLab^®^; Scientific Formosa, Kaohsiung, Taiwan), and Microsoft Excel (part of the Microsoft 365 suite) (Microsoft, Redmond, WA, USA) with the “Solver” add-in function. 

The experimental data obtained from the experiments comprised various types of ionic currents and were presented using the mean value with corresponding standard error of the mean (SEM). The sample size (n) denoted the number of cells from which data were gathered, and the SEM error bars were included in the plots. The Kolmogorov–Smirnov test for normality suggested that the data distribution was acceptable and could be assumed as normal. Paired or unpaired Student’s *t*-tests between the two groups were applied. To assess differences between more than two groups, we used an analysis of variance (ANOVA-1 or ANOVA-2) with or without repeated measures, followed by a post-hoc Fisher’s least significant difference test. Differences were considered statistically significant at a *p* value < 0.05 (* or ** in the figures denote significance).

## 3. Results

### 3.1. Effect of Cannabidiol (CBD) on the M-Type K^+^ Current (I_K(M)_) Identified in Pituitary GH_3_ Cells

In the initial whole-cell current recordings, we first investigated whether CBD had an effect on the density of *I*_K(M)_ in these cells. To assess whether CBD caused any perturbations in *I*_K(M)_, we exposed the cells to a high-K^+^, Ca^2+^-free solution containing 1 μM tetrodotoxin (TTX). The measuring electrode was filled with a K^+^-enriched solution, and we then held the examined cell at the level of −50 mV, and a 1 s depolarizing pulse to −10 mV was applied to evoke *I*_K(M)_. Under the experimental conditions described above, the activation of *I*_K(M)_ was indicated by a slowly activating time course in response to long-lasting step depolarization. As the reversal potential of K^+^ ions was around 0 mV, the resulting *I*_K(M)_ was a slowly activating inward K^+^ current that directed the flow of K^+^ ions into the interior of the cell [11,12]. Of interest, we observed a progressive reduction in the density of *I*_K(M)_ following a 1 s membrane depolarization from −50 to −10 mV. This reduction was observed one minute after cells were subjected to the increasing concentrations of CBD (Figure 1A). For instance, when cells were exposed to 1 or 3 μM CBD, the density of *I*_K(M)_ measured at the end of a 1 s membrane depolarization decreased to 3.8 ± 0.4 pA/pF (*n* = 8, *p* < 0.05) or 1.9 ± 0.2 pA/pF (*n* = 8, *p* < 0.05), respectively, from a control value of 3.9 ± 0.4 pA/pF (*n* = 8). Following the cessation of CBD exposure and subsequent washout, the current density returned to 3.7 ± 0.3 pA/pF (*n* = 8). Additionally, Figure 1B displays the concentration–response curve for the CBD-induced inhibition of the density of *I*_K(M)_, which was constructed with an IC_50_ value of 3.6 μM. The results reflect that exposing GH_3_ cells to CBD can cause a concentration-dependent decrease in the density of *I*_K(M)_, indicating a depressant action.

### 3.2. Effect of CBD on the Current Density Versus Voltage Relationship and the Quasi-Steady-State Activation Curve of I_K(M)_ in GH_3_ Cells

To further investigate the inhibitory effect of CBD on *I*_K(M)_, we analyzed the current density versus voltage relationship of *I*_K(M)_ with and without CBD addition. Figure 2A illustrates the quasi-steady-state current density versus voltage relationship of *I*_K(M)_ obtained during the control period and after exposure to 3 μM CBD in these cells. We observed a marked decrease in the density of *I*_K(M)_ elicited by depolarizing steps, particularly at voltage levels ranging between −30 and −10 mV, upon the addition of CBD. 

Moreover, the quasi-steady-state activation curve of *I*_K(M)_ in the absence and presence of 3 μM CBD was constructed and plotted in Figure 2B. The normalized conductance of *I*_K(M)_ with or without the presence of CBD was fitted to a Boltzmann function using the method described in Section 2. In the control, *V*_1/2_ = −17.5 ± 0.8 mV, *k* = 4.9 ± 0.5 (*n* = 7), while in the presence of 3 μM CBD, *V*_1/2_ = −10.4 ± 0.8 mV, *k* = 4.9 ± 0.5 (*n* = 7). The data reflected that CBD not only decreased the maximal magnitude of *I*_K(M)_, but it also shifted the quasi-steady-state activation curve of *I*_K(M)_ to more depolarized potential (i.e., in the rightward direction) by approximately 7 mV. However, we found no clear adjustments in the slope factor (*k* value or steepness) of the curve when exposed to CBD, suggesting that there was no change in the gating charge of the curve during its exposure. Alternatively, the CBD-induced inhibition of *I*_K(M)_ in unclamped cells could depend on the pre-existing resting membrane. 

### 3.3. Comparisons among Effects of CBD, CBD plus Naloxone, Linopirdine (Lino), Thyrotropin-Releasing Hormone (TRH), and Liraglutinide (Lira) on the Density of I_K(M)_ in GH_3_ Cells

In another series of whole-cell experiments, we tested the possible effects of CBD, CBD plus naloxone, Lino, TRH, and Lira on the density of *I*_K(M)_ elicited by membrane depolarization from −50 to −10 mV. Lino has been demonstrated to inhibit *I*_K(M)_ potently [27], while TRH can suppress the magnitude of *I*_K(M)_ [11]. Lira was known to be an agonist of glucagon-like peptide-1 (GLP-1) receptor [28,29]. As demonstrated in Figure 3A,B, CBD at a concentration of 3 μM produced an inhibitory effect on the *I*_K(M)_ density; however, a subsequent addition of 10 μM naloxone failed to counteract CBD-induced inhibition of *I*_K(M)_. Meanwhile, the presence of Lino (3 μM), TRH (1 μM), or Lira (1 μM) was effective at suppressing the magnitude of *I*_K(M)_ (Figure 3B). Naloxone is known to block opioid receptors. Naloxone at a concentration of 30 μM also did not cause any effect on the CBD-induced decrease in *I*_K(M)_. Hence, similar to TRH, Lira can suppress *I*_K(M)_ possibly through its binding to GLP-1 receptors in pituitary cells [28]. However, further addition of naloxone exerted no effect on the CBD-mediated decrease in *I*_K(M)_, suggesting that the CBD effect on *I*_K(M)_ is independent of binding to opioid receptors.

### 3.4. Modification by CBD of I_K(M)_ Elicited by Pulse-Train (PT) Stimulation

Previous work has shown the capability of *I*_K(M)_ to maintain the availability of voltage-gated Na^+^ (Na_V_) channels during prolonged high-frequency firing [16]. Therefore, we further examined whether the CBD presence may modify the extent of *I*_K(M)_ during PT depolarizing stimuli from −50 to −10 mV in GH_3_ cells. As depicted in Figure 4, a one-minute exposure to 3 μM CBD resulted in a reduction in the density of both activating and deactivating *I*_K(M)_ during 1 s PT stimulation. For instance, the presence of CBD (3 μM) decreased the density of activating *I*_K(M)_ from 7.0 ± 1.1 to 3.7 ± 0.6 pA/pF (*n* = 7, *p* < 0.05) and the density of deactivating *I*_K(M)_ from 25.8 ± 3.8 to 7.7 ± 1.2 pA/pF (*n* = 7, *p* < 0.05). Subsequent to the washout of the CBD, the densities of activating and deactivating *I*_K(M)_ were restored to 6.8 ± 0.9 (*n* = 6) and 24.9 ± 3.5 pA/pF (*n* = 6), respectively. The obtained data provided insights into the persistent effectiveness of CBD-induced inhibition of *I*_K(M)_ even under conditions of high PT stimulation. Thus, it is plausible to consider that the presence of Na_V_ channels during high-frequency firing in unclamped excitable cells may experience additional inhibition during CBD exposure, despite CBD’s previously descried potential to suppress Na_V_ channel activity [8].

### 3.5. Mild Inhibitory Effect of CBD on the erg-Mediated K^+^ Current (I_K(erg)_) in GH_3_ Cells

We continued by exploring whether CBD can exert any effect on another type of K^+^ currents (i.e., *I*_K(erg)_) enriched in these cells. Cells were kept in high-K^+^, Ca^2+^-free solution and we filled up the measuring electrode with K^+^-enriched solution. To evoke *I*_K(erg)_, we held the tested cell at the level of −10 mV and a hyperpolarizing step to −90 mV for a duration of 1 s was applied to it. As shown in Figure 5, one minute after being exposed to CBD (10 μM), the density of *I*_K(erg)_ in response to membrane hyperpolarization was significantly decreased to 8.6 ± 1.1 pA/pF (*n* = 8, *p* < 0.05) from a control value of 11.2 ± 1.5 pA/pF (*n* = 8), while CBD at a concentration of 3 μM did not have a clear effect on *I*_K(erg)_ density elicited by membrane hyperpolarization. Following the removal of the 10 μM CBD and subsequent washout, the *I*_K(erg)_ density returned to 11.0 ± 1.4 pA/pF (*n* = 8). Moreover, during continued exposure to 10 μM CBD, a further addition of NS1643 (10 μM) was capable of attenuating CBD-mediated inhibition of *I*_K(erg)_ (10.9 ± 1.5 pA/pF, *n* = 8, *p* < 0.05) in these cells. NS1643 has been reported to stimulate *I*_K(erg)_ [30]. The results therefore indicate that exposure to CBD slightly suppressed the density of *I*_K(erg)_ in GH_3_ cells.

### 3.6. Failure of CBD Effect on Voltage-Gated Na^+^ Current (I_Na_) in GH_3_ Cells

CBD was recently reported to suppress the magnitude of Na_V_1.4-encoded *I*_Na_ [8]. In our study, we conducted additional experiments to investigate whether the presence of CBD can also modify *I*_Na_ in GH_3_ cells. As shown in Figure 6, in whole-cell current recordings, the magnitude of *I*_Na_ elicited in response to a short depolarizing pulse to −10 from a holding potential of −100 mV was robustly observed; however, the *I*_Na_ density remained unaltered during a 2 min continued exposure to 10 μM CBD. Similarly, cannabichromene, a compound known to bind to cannabinoid receptor 1 [1], did not have any effect on the density of *I*_Na_. Moreover, with continued presence of 10 μM CBD, the further addition of ranolazine (Ran, 10 μM) was effective at suppressing *I*_Na_ density, while the addition of either tefluthrin (Tef, 10 μM) or telmisartan (TEL, 10 μM) enhanced current density. Ran is an inhibitor of late *I*_Na_ [31], while Tef or TEL can stimulate *I*_Na_ effectively [32]. Therefore, unlike Na_V_1.4-encoded currents [8], the *I*_Na_ observed in GH_3_ cells appears to be resistant to modulation by CBD.

### 3.7. Effect of CBD on Hyperpolarization-Activated Cation Current (I_h_) Measured in GH_3_ Cells

We continued to examine the perturbations caused by CBD in *I*_h_ in these cells. To investigate the effects of CBD on *I*_h_, we performed experiments under specific conditions. First, we placed the cells in Ca^2+^-free Tyrode’s solution containing 0.5 mM CdCl_2_ and 1 μM TTX. We used CdCl_2_ and TTX to block voltage-gated Ca^2+^ and Na^+^ currents, respectively. The measuring electrode was backfilled with K^+^-enriched internal solution to evoke *I*_h_. As the whole-cell mode was established, the tested cell was maintained at the level of −40 mV in voltage-clamp mode, and the voltage steps were subsequently applied (2 s in duration) to a series of voltages ranging between −110 and −20 mV in 10 mV increments. As demonstrated in Figure 7A, one minute after cells were exposed to 3 μM CBD, the *I*_h_ density elicited by 2 s membrane hyperpolarization to −110 mV from a holding potential of −40 mV was decreased. However, even with continued exposure to 3 μM CBD, the subsequent addition of 10 μM naloxone failed to reverse the CBD-induced decrease in *I*_h_ density. Moreover, the quasi-steady-state current remained largely unaltered during CBD exposure.

Of interest, one minute after cell exposure to 3 μM CBD, the density of *I*_h_ evoked by the 2 s long step hyperpolarization progressively decreased (Figure 7B,C). For instance, at the levels of −100 and −110 mV, it was observed that following a one-minute exposure to 3 μM CBD, the density of *I*_h_ diminished to 4.6 ± 0.8 pA/pF (*n* = 7, *p* < 0.05) and 7.6 ± 1.1 pA/pF (*n* = 7, *p* < 0.05), respectively. These values were reduced from the control values of 9.7 ± 1.4 pA/pF (*n* = 7) and 15.6 ± 1.7 pA/pF (*n* = 7). When CBD was removed, the *I*_h_ density at −100 and −110 mV returned to 9.3 ± 1.1 pA/pF (*n* = 7) and 15.2 ± 1.6 pA/pF (*n* = 7), respectively. Additionally, the activating time constant (τ_act_) of *I*_h_ activated by the 2 s hyperpolarizing step from −40 to −110 mV decreased to 1.81 ± 0.04 s (*n* = 7, *p* < 0.05) from a control value of 1.07 ± 0.03 s (*n* =7). Moreover, with continued exposure to 3 μM CBD, a further addition of naloxone (10 μM) failed to reverse the CBD-mediated decrease in *I*_h_ density.

Figure 7D illustrates the quasi-steady-state activation curve of *I*_h_ acquired in the control period and during exposure to 3 μM CBD. According to the least-squares minimization procedure, the best fitting parameters (i.e., *V*_1/2_ and *k*) for such a steady-state activation curve of the current were convergently acquired with or without the addition of CBD. In the presence of 3 μM CBD, the value of *V*_1/2_ for the activation curve of the current was −101 ± 4 mV (*n* = 7), which was significantly distinguishable from that in the control period, −89 ± 4 mV (*n* = 7, *p* < 0.05). However, the *k* values in the absence (7.2 ± 0.3, *n* = 7) and presence of 3 μM CBD (7.3 ± 0.3, *n* = 7, *p* > 0.05) did not differ significantly. The experimental results showed that during exposure to 3 μM CBD, the activation curve of *I*_h_ was shifted toward more hyperpolarized potential with no change in the slope factor of the curve (i.e., the steepness of the activation curve). Therefore, it appears likely that no change in the gating charge of *I*_h_ activation curve was altered by CBD exposure. The responsiveness of *I*_h_ to CBD is influenced by varying levels of pre-existing resting membrane potential.

As illustrated in Figure 7E, the presence of various concentrations of CBD can suppress the density of *I*_h_ in a concentration-dependent manner. By virtue of a non-linear least-squares fit to the experimental data, the IC_50_ value for the inhibitory effect of CBD on *I*_h_ was 3.3 μM, a value that is similar to that needed to suppress *I*_K(M)_ density described above. These results indicate that CBD can exercise a depressant action on *I*_h_ elicited by membrane hyperpolarization in these cells.

### 3.8. Effect of CBD on the Hysteretic Behavior (i.e., Voltage-Dependent Hysteresis (Hys_(V)_)) of I_h_ Elicited by Isosceles-Triangular Ramp Voltage (V_ramp_) in GH_3_ Cells

Previous work has demonstrated the effectiveness of *I*_h_’s Hys_(V)_ strength in affecting either various patterns of bursting firing or action potential configuration in varying types of excitable cells [33,34]. Therefore, we continued by determining whether and how the presence of CBD may adjust the *I*_h_ strength activated in response to long-lasting triangular V_ramp_. In this series of experiments, in the control period (i.e., absence of CBD), we maintained the tested cell at the level of −40 mV, and an upsloping (forward) limb from −150 to −40 mV followed by a downsloping (backward) limb back to −150 mV (i.e., upright isosceles-triangular V_ramp_) for a total duration of 2 s with a ramp speed of ±55 mV/s was thereafter applied to evoke *I*_h_’s hysteresis (Hys_(V)_) (Figure 8A,B). In accordance with previous studies [21,33,34], the Hys_(V)_ of *I*_h_ in response to such double V_ramp_ was robustly observed and sensitive to suppression by ivabradine (IVA), an inhibitor of *I*_h_ [19,20,26]. Of additional interest, upon cell exposure to 3 μM CBD, the strength of *I*_h_’s Hys_(V)_ responding to both rising and falling limbs of double V_ramp_ progressively became depressed (Figure 8C). For instance, as the triangular V_ramp_ was applied, the value of ∆area (i.e., the difference in area enclosed by the curve in the forward and backward direction) for Hys_(V)_ in the control period was 248 ± 52 mV·(pA/pF) (*n* = 7), while the ∆area value of *I*_h_ in the presence of 3 μM CBD was significantly reduced to 142 ± 28 mV·(pA/pF) (*n* = 7, *p* < 0.05). Moreover, a subsequent addition of oxaliplatin (OXAL, 10 μM) effectively reversed the CBD-induced decrease in Hys_(V)_’s strength. OXAL has been reported to activate *I*_h_ [25,35]. Therefore, the results indicate that cell exposure to CBD can modify the magnitude and gating properties of *I*_h_ observed in GH_3_ cells.

## 4. Discussion

This study identified seven significant findings. (1) Cannabinoid (CBD) exposure resulted in a concentration-dependent suppression of M-type K^+^ current (*I*_K(M)_) in pituitary GH_3_ cells, with an IC_50_ of 3.6 μM. (2) The presence of CBD caused a rightward shift in the quasi-steady-state activation curve of *I*_K(M)_ without changes in the slope factor (i.e., *k* value) of the curve. (3) The CBD-induced block of *I*_K(M)_ was not reversible by further addition of naloxone. (4) CBD exposure also suppressed *I*_K(M)_ elicited by pulse-train (PT) depolarizing stimuli. (5) CBD slightly reduced the magnitude of *erg*-mediated K^+^ current (*I*_K(erg)_) but did not affect voltage-gated Na^+^ current (*I*_Na_) in GH_3_ cells. (6) Cell exposure to CBD resulted in the inhibition of hyperpolarization-activated cation current (*I*_h_) with an IC_50_ of 3.3 μM. (7) The quasi-steady-state activation curve of *I*_h_ was shifted to a more hyperpolarized potential without changes in the curve’s slope factor. Based on the findings of this study, it can be inferred that CBD exerts a significant impact on transmembrane ionic currents, particularly *I*_K(M)_ and *I*_h_. Assuming that these effects would occur in vivo, they would likely influence the firing frequency of action potential generation in cells. Therefore, further research is necessary to determine the possible mechanisms of action of CBD or other structurally similar compounds in the body, such as the anxiolytic effects [3,6,7].

Different to some extent from a previous study [17], the current study found that the presence of CBD could suppress the magnitude of *I*_K(M)_. The IC_50_ value for CBD to inhibit the density of *I*_K(M)_ in GH_3_ cells was 3.6 μM. The quasi-steady-state activation curve of *I*_K(M)_ in GH_3_ cells shifted towards a more depolarized potential, without changes in the curve’s steepness. Additionally, the CBD presence could suppress the *I*_K(M)_ elicited by PT stimulation. These findings suggest that the responsiveness of *I*_K(M)_ in unclamped cells is influenced by various confounding factors, including the level of the pre-existing resting membrane potential, CBD concentration, patterns of action potential firing, or combinations of these variables.

It needs to be mentioned that, concerning the steady-state activation curve of *I*_K(M)_, it is important to have a sufficiently long step duration to accurately assess the slope factor and V_1/2_ values. This is essential for a more precise determination of the activation curve of this current, followed by the application of the Boltzmann equation to deduce V_1/2_ and the slope factor, subsequently allowing to predict the gating charge value. However, in the cells we have investigated (such as GH_3_ cells), when we employed a voltage-clamp protocol that exceeds 1 sec, we often observed the appearance of the inactivation process from other types of K^+^ currents. This phenomenon, to a certain extent, led to confusion with the inactivation of other delayed-rectifier K^+^ currents. The primary reason behind this is that GH_3_ cells, in addition to *I*_K(M)_, still possess various types of delayed-rectifier K^+^ currents. This is a common issue encountered when conducting experiments on native excitable cells. Using cells with KCNQx-encoded currents may help mitigate these complications. Nonetheless, the exposure of CBD to GH_3_ cells did suppress the strength of *I*_K(M)_ and shift the quasi-steady-state activation curve of *I*_K(M)_ to a more depolarized potential without affecting the curve’s slope factor.

Previous research has shown that a train of depolarizing pulses can effectively alter the magnitude of *I*_Na_, which is a current that decays exponentially over time [16]. More recently, it has been demonstrated that the magnitude of *I*_K(M)_ can regulate the availability of Na_V_ channels during prolonged high-frequency firing [16]. In our study, we found that exposure of GH_3_ cells to CBD suppressed the magnitude of *I*_K(M)_ during PT depolarizing stimuli. As a result, the availability of Na_V_ channels during sustained high-frequency firing may be reduced, leading to a decrease in reliable presynaptic spiking and synaptic transmission at high frequencies [9,10,14].

Our study observed that CBD did not significantly alter the magnitude of *I*_Na_ in GH_3_ cells, nor did cannabichromene, an agonist of cannabinoid receptor 1 [1]. These results appear to differ from the findings of a previous study conducted by Huang et al. [8]. They reported the suppression of the density of Na_V_1.4 currents by CBD. The reason for this discrepancy is not yet understood; however, it is possible that Na_V_1.4 currents affected by CBD [8] are primarily expressed in skeletal muscle cells rather than endocrine or neuroendocrine cells. It is thus pertinent to conduct further investigation to determine if the effects of CBD on *I*_Na_ are specific to certain tissues.

Four mammalian isoforms of HCN, which are HCN1, HCN2, HCN3, and HCN4, make up the macroscopic *I*_h_ (or *I*_f_) [22,33]. Among them, HCN2, HCN3, and a combination of HCN2 and HCN3 channels are highly expressed in GH_3_ cells and other types of endocrine or neuroendocrine cells [36]. In our study, we observed that CBD caused a suppressive effect on *I*_h_ in a concentration- and voltage-dependent manner. The IC_50_ value for CBD to suppress *I*_h_ was estimated as 3.3 μM, and the quasi-steady-state activation curve of *I*_h_ was shifted to a more hyperpolarizing potential with no changes in the curve’s steepness. The CBD exposure also resulted in a considerable reduction in the strength of *I*_h_’s Hys_(V)_ elicited by long-lasting triangular V_ramp_. The effect of CBD on this Hys_(V)_ behavior would be linked to its effect on the gating mechanism of HCN channels [37,38]. Therefore, apart from its effectiveness in inhibiting *I*_K(M)_, as detailed above, CBD can also induce changes in the magnitude, gating kinetics, and Hys_(V)_ behavior of *I*_h_ in a manner that may have pharmacological, therapeutic, or toxicological significance. It is worth noting that clinically achievable levels of CBD have been reported [39].

It is important to mention that, as CBD affects ion channels on the cell membrane with similar potency, it appears unlikely that these effects are solely mediated by CBD binding to cannabinoid receptors. If these effects were due largely to binding to cannabinoid receptors, the specificity of the effects should be particularly evident. Indeed, the subsequent addition of naloxone, an antagonist of opioid receptors, was not found to reverse CBD-induced inhibition of *I*_K(M)_, despite the ability of CBD to modulate the activity of μ- and δ-opioid receptors [4,40]. Therefore, CBD’s influence on these currents is likely to be rapid and direct, rather than dependent on binding to cannabinoid or opioid receptors. Our findings offer substantial promise in terms of utilizing CBD in fundamental neuroscience and clinical research. As for how CBD modulates ionic currents and potentially disrupts specific downstream signaling pathways, further in-depth research is warranted in the future.

## Figures and Tables

**Figure 1 biomedicines-11-02651-f001:**
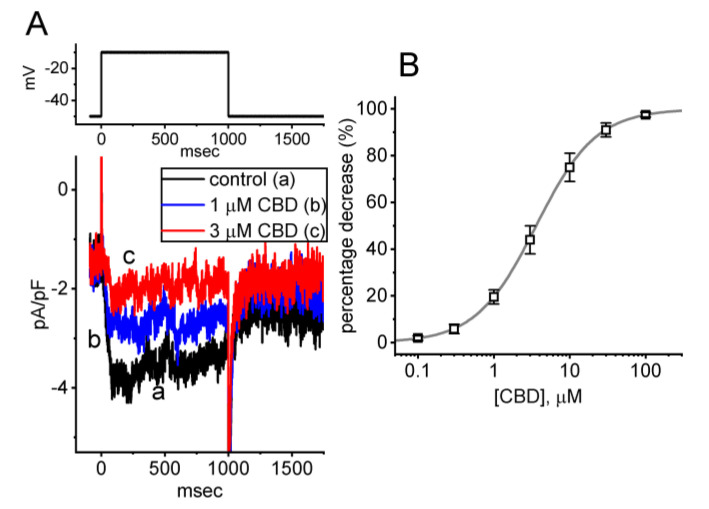
Effect of cannabidiol (CBD) on M-type K^+^ current (*I*_K(M)_) identified in pituitary GH_3_ cells. In this set of experiments, we bathed cells in high-K^+^, Ca^2+^-free solution, which contained 1 μM tetrodotoxin (TTX), and the measuring electrode used was filled up with a K^+^-enriched solution. (**A**) Representative current traces acquired in the control period (a) (i.e., absence of CBD) and during cell exposure to 1 μM CBD (b) or 3 μM CBD (c). The upper part indicates the voltage-clamp protocol applied. (**B**) Concentration–response curve of CBD-induced block of *I*_K(M)_ density observed in GH_3_ cells (mean ± SEM; *n* = 8 for each point). The sigmoidal line drawn represents the goodness-of-fit to the Hill equation, as described in Section 2. The IC_50_ values for the CBD-mediated inhibition of *I*_K(M)_ was optimally estimated to be 3.6 μM.

**Figure 2 biomedicines-11-02651-f002:**
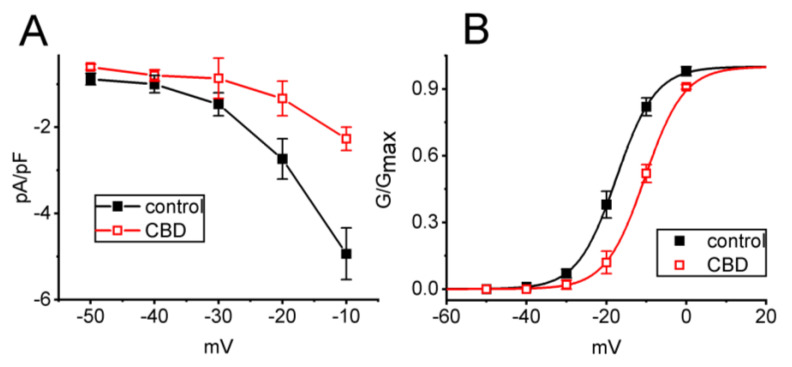
Effect of CBD on mean current density versus voltage relationship (**A**) and the quasi-steady-state activation curve (**B**) of *I*_K(M)_. The experimental procedures employed in these experiments were similar to those outlined in Figure 1. The examined cell was maintained at −50 mV and a series of command voltages ranging from −50 to 0 mV in 10 mV steps were applied to it. (**A**) Mean current density versus voltage relationship of *I*_K(M)_ in the absence (black filled squares) and presence (red open squares) of 3 μM CBD (mean ± SEM; *n* = 7 for each point). Current density was measured at the end of each voltage step. (**B**) Quasi-steady-state activation curve of *I*_K(M)_ in the control (black filled squares) and during exposure to 3 μM CBD (red open squares) (mean ± SEM; *n* = 7 for each point). The smooth lines described in Section 2 were optimally generated by fitting the Boltzmann equation for the activation curve of the current using a least-squares method. The statistical analyses in (**A**) and (**B**) were undertaken by ANOVA-2 for repeated measures, *p* (factor 1, groups among data taken at different levels of membrane potential) < 0.05, *p* (factor 2, groups between the absence and presence of CBD) < 0.05, *p* (interaction) < 0.05, followed by post-hoc Fisher’s test, *p* < 0.05. Of note, exposure to 3 μM CBD resulted in a rightward shift of the activation curve of *I*_K(M)_ with no change in the slope factor of the curve.

**Figure 3 biomedicines-11-02651-f003:**
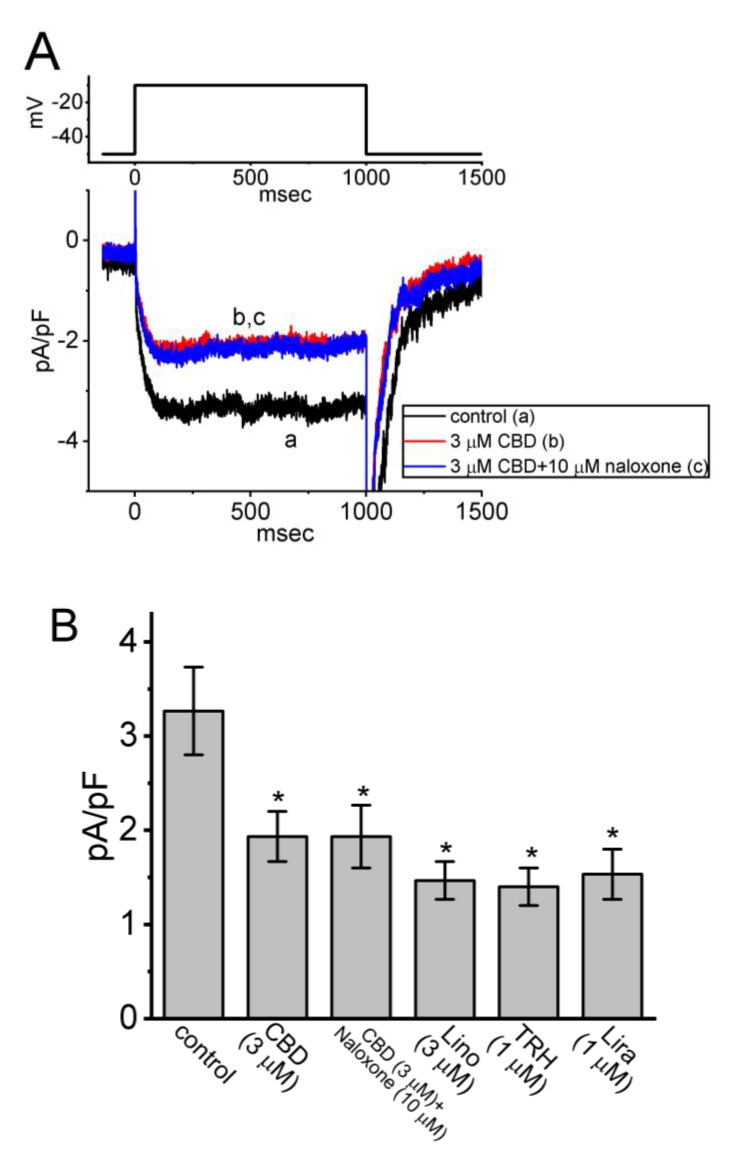
Comparison among the impact of CBD, CBD plus naloxone, linopirdine (Lino), thyrotropin-releasing hormone (TRH), and liraglutinide (Lira) on the observed density of *I*_K(M)_ in GH_3_ cells. (**A**) Representative current traces acquired in the control period (a) and during exposure to 3 μM CBD (b) or 3 μM CBD plus 10 μM naloxone (c). The upper part shows the voltage-clamp protocol applied. (**B**) Summary bar graph showing effect of CBD, CBD plus naloxone, linopirdine, thyrotropin-releasing hormone, and liraglutinide on *I*_K(M)_ in GH_3_ cells. Current density was measured at the end of 1 s depolarizing pulse from −50 to −10 mV. Each bar indicates the mean ± SEM (*n* = 7). Data analysis was performed by ANOVA-1 (*p* < 0.05). * Significantly different from control (*p* < 0.05).

**Figure 4 biomedicines-11-02651-f004:**
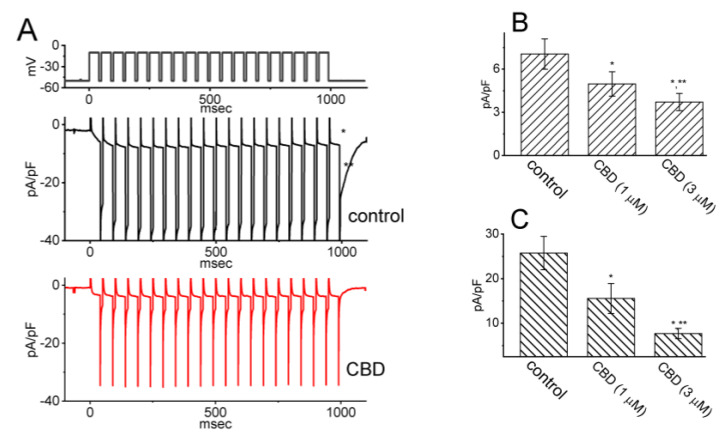
Impact of CBD on the activation of *I*_K(M)_ through pulse-train (PT) stimulation in GH_3_ cells. To conduct the experiment, cells were placed in a high-K^+^, Ca^2+^-free solution, and the PT stimulation protocol involved a series of 40 depolarizing pulses lasting 20 ms each, applied at −10 mV with 5 ms intervals, for a total duration of 1 s. (**A**) Representative current traces are presented, depicting recordings acquired during the control period (upper trace in black) and in the presence of 3 μM CBD (lower trace in red). The top portion of the figure shows the applied voltage-clamp protocol. The * symbol indicates the activating *I*_K(M)_, while ** represents the deactivating (or tail) component of *I*_K(M)_ obtained after returning to −50 mV. Summary bar graphs in (**B**,**C**) display the activating and deactivating densities of *I*_K(M)_, respectively, in the absence and presence of 1 or 3 μM CBD. The values are presented as mean ± SEM, with each bar representing data from seven independent experiments. The activating density of *I*_K(M)_ was measured at the end of the PT depolarizing pulses from −50 to −10 mV, while the deactivating density was measured following the return to −50 mV. Data analyses in (B) and (C) were performed by ANOVA-1 (*p* < 0.05). The * symbol indicates statistical significance when compared to the control group (*p* < 0.05), while ** denotes statistical significance when compared to the CBD (1 μM) alone group (*p* < 0.05).

**Figure 5 biomedicines-11-02651-f005:**
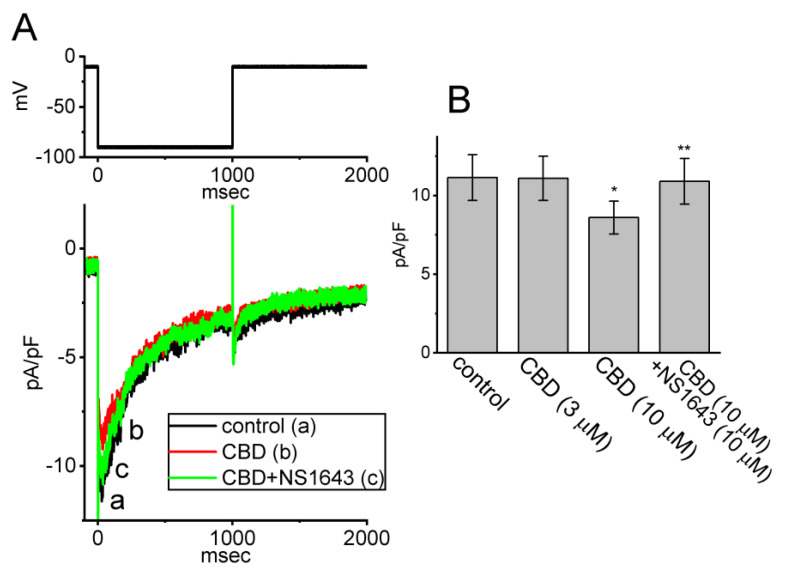
Mild inhibitory effect of CBD on *erg*-mediated K^+^ current (*I*_K(erg)_) in GH_3_ cells. We placed cells in high-K^+^, Ca^2+^-free solution containing 1 μM TTX, and the measuring pipette was filled with a K^+^-enriched solution. (**A**) Representative current traces obtained in the control period (a, black color), and during cell exposure to 10 μM CBD (b, red color) or to 10 μM CBD plus 10 μM NS1643 (c, green color). The voltage protocol applied is illustrated in the upper part. (**B**) Summary bar graph demonstrating effects of CBD (3 or 10 μM) and 10 μM CBD plus 10 μM NS1643 on the density of *I*_K(erg)_ (mean ± SEM; *n* = 8 for each bar). Current density (i.e., deactivating *I*_K(erg)_) was measured at the beginning of 1 s step hyperpolarization from −10 to −90 mV. Data analysis was performed by ANOVA-1 (*p* < 0.05). * Significantly different from control (*p* < 0.05) and ** significantly different from CBD (10 μM) alone group (*p* < 0.05).

**Figure 6 biomedicines-11-02651-f006:**
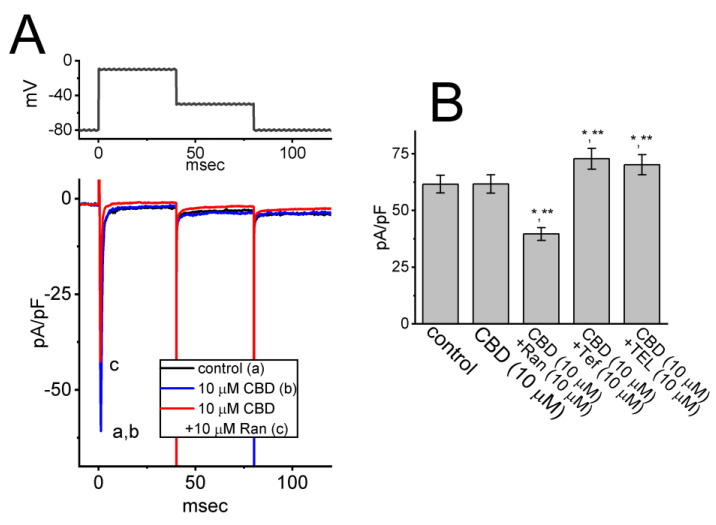
Inability of CBD to modify the density of voltage-gated Na^+^ current (*I*_Na_) in GH_3_ cells. In these experiments, we placed cells in Ca^2+^-free Tyrode’s solution that contained 10 mM tetraethylammonium chloride, and the recording electrode was filled with a Cs^+^-enriched solution. (**A**) Representative current traces acquired in the control period (a, black color) and during cell exposure to either 10 μM CBD alone (b, blue color) or 10 μM CBD plus 10 μM ranolazine (Ran) (c, red color). The voltage-clamp protocol applied is indicated in the upper part. (**B**) Summary bar graph demonstrating effects of CBD, CBD plus ranolazine, CBD plus tefluthrin (Tef), and CBD plus telmisartan (TEL) on the peak density of *I*_Na_ (mean ± SEM; *n* = 7 for each bar). Of note, the presence of CBD (10 μM) did not cause any effect on the peak *I*_Na_; however, further addition of Ran decreased *I*_Na_ density, while that of either Tef or TEL increased it. Data analysis was performed by ANOVA-1 (*p* < 0.05). * Significantly different from control (*p* < 0.05) and ** significantly different from CBD (10 μM) alone group (*p* < 0.05).

**Figure 7 biomedicines-11-02651-f007:**
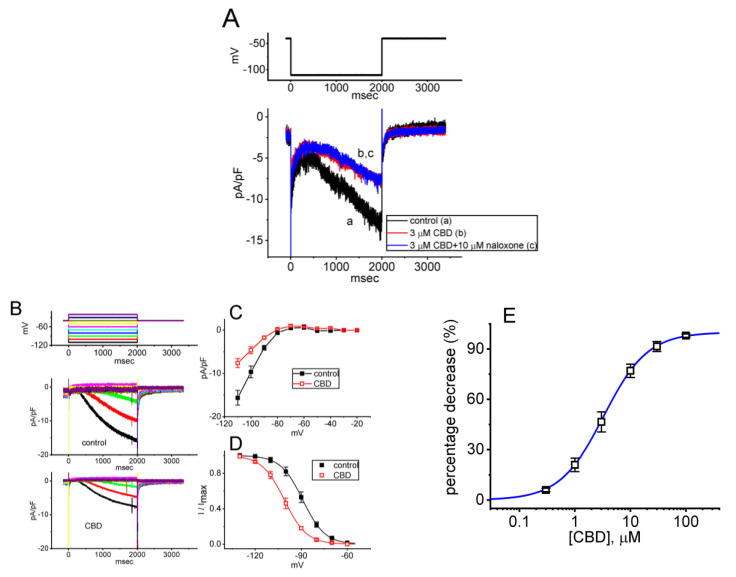
Inhibitory effect of CBD on mean current density versus voltage relationship of hyperpolarization-activated cation current (*I*_h_) recorded in GH_3_ cells. In this set of measurements, cells were kept in Ca^2+^-free Tyrode’s solution that contained 1 μM TTX, and the measuring electrode was filled up with a K^+^-enriched solution. (**A**) Representative current traces acquired in the control period (a) and during exposure to 3 μM CBD (b) or 3 μM CBD plus 10 μM naloxone (c). The upper part indicates the voltage-clamp protocol used. (**B**) Representative current traces obtained in the control period (upper part) and during the presence of 3 μM CBD (lower part). The uppermost part represents the voltage-clamp protocol used, while the different colors correspond to different evoked currents (shown in the figure below). (**C**) Mean current density versus voltage relationship of *I*_h_ acquired in the absence (black filled squares) and presence (red open squares) of *I*_h_ density (mean ± SEM; *n* = 7 for each point). Current density was measured at the end of each hyperpolarizing pulse for a duration of 2 s. (**D**) Quasi-steady-state activation curve of *I*_h_ obtained in the control period (black filled squares) and during exposure to 3 μM CBD (red open squares) (mean ± SEM; *n* = 7 for each point). The sigmoid curve in the figure represents the Boltzmann equation (shown in Section 2), which was fit to the data using least-squares minimization. The statistical analyses for (**C**,**D**) were undertaken by ANOVA-2 for repeated measures, *p* (factor 1, groups among data taken at different levels of membrane potential) < 0.05, *p* (factor 2, groups between the absence and presence of CBD) < 0.05, *p* (interaction) < 0.05, followed by post-hoc Fisher’s test, *p* < 0.05. (**E**) Concentration–response curve of CBD-induced inhibition of *I*_h_ density (mean ± SEM; *n* = 8 for each point). Current densities in the presence of different CBD concentrations were taken at the end of a 2 s hyperpolarizing pulse from −40 to −110 mV. The smooth curve was generated using the least-squares method and fitted to the Hill equation, as detailed in Section 2.

**Figure 8 biomedicines-11-02651-f008:**
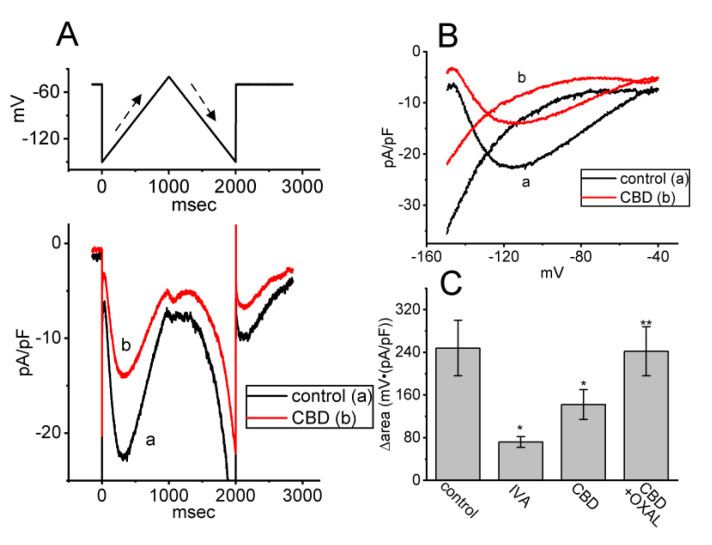
Impact of CBD on *I*_h_ elicited by long-lasting triangular ramp voltage (V_ramp_) in GH_3_ cells. (**A**) Representative current traces acquired without (a, black color) or with (b, red color) the presence of 3 μM CBD. The upper part indicates the V_ramp_ protocol applied, while dashed arrows show the ascending and descending limbs of V_ramp_ as a function of time. (**B**) Representative current versus voltage relationship (i.e., voltage-dependent hysteresis, Hys_(V)_) of *I*_h_ elicited by 2 s triangular V_ramp_ with or without the addition of 3 μM CBD. The dashed arrow indicates current trajectory which passes over time during triangular V_ramp_. (**C**) Summary bar graph showing effects of ivabradine (IVA, 3 μM), CBD (3 μM), and CBD (3 μM) plus oxaliplatin (OXAL, 10 μM) on the area (Δarea) of V_ramp_-induced Hys_(V)_ in *I*_h_ (mean ± SEM; *n* = 7 for each bar). The Δarea represents the area enclosed by the upward (ascending) and downward (descending) limbs of the *I*_h_ elicited using an isosceles triangular V_ramp_ usually ranging between −50 and −130 mV. Data analysis was performed by ANOVA-1 (*p* < 0.05). * Significantly different from control (*p* < 0.05) and ** significantly different from CBD (3 μM) alone group (*p* < 0.05).

## Data Availability

Data was available by request.

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
