# Peer review of "Cannabidiol Modulates M-Type K+ and Hyperpolarization-Activated Cation Currents"

_biomedicines, 2023, doi:10.3390/biomedicines11102651_

Round 1

Reviewer 1 Report

The manuscript titled: “The efficacy of cannabidiol (CBD) in inhibiting M-type K+ and hyperpolarization-activated cation currents” aimed to investigate how CBD modifies various types of ionic currents in pituitary GH3 cells. The functional analysis revealed the inhibition of M-type K+ currents by the tested compound in micromolar range with concentration-dependent manner and shifting the steady-state of activation to a more depolarized potential. Such modification is not mediated by opioid receptors. Also, results showed that the magnitude of ERG-mediated K+ currents was slightly reduced by adding 10 mM CBD, without affecting the Nav currents. Furthermore, CBD decreased the magnitude of hyperpolarization-activated cation currents and shifting their steady-state of activation to the leftward direction. These findings shed the light on the significant impact on the CBD’s modification of ionic currents in excitable cells.

Several Major and minor points to be address in the current manuscript:

Major:

- Do the authors examined the specific type of Kv channels that might be inhibited by the CBD besides ERG channels, using heterologous expression system, such as monoclonal stable cell-lines?

 - How do you describe the discrepancy between your results and that of reference 8? lanes 249-252 should be clarified and likewise in the discussion with exploring the CBD-sensitive subtypes of Nav channels.

- In Figure 3, no current trances were shown to support the error-bar diagram.

- In a conclusion remark, authors might show more potentials of their finding on exploiting CBD in basic neuroscience and clinical research.

Minor:

- In many places within the test, the references are not numbered, which is not adhere to the guidelines of the journal. For example, lane 236.

- Also, a typo is shown all over the text that m symbol is replaced by another one.

In Figure 4 B and C, panels are small and disproportional to panel A.

- In Figure 6 B, the panel is disproportional and unread comparing to panel A.

The quality of English matches to the guidelines and level of the journal. 

Author Response

Ans: Thanks for the valuable and insightful comments provided by the reviewer.

Several Major and minor points to be address in the current manuscript:

Major:

The following response was provided in addressing the reviewer’s comments in a point-by-point manner.

- Do the authors examined the specific type of Kv channels that might be inhibited by the CBD besides ERG channels, using heterologous expression system, such as monoclonal stable cell-lines?

Ans: Thanks for the reviewer’s comments. In our study, we did not use ERG-expressing monoclonal stable cell lines. However, this will be important to be tested in the near future. The primary cell type utilized in this study is the GH3 cell line, derived from the anterior pituitary somatolactotrophs.

 - How do you describe the discrepancy between your results and that of reference 8? lanes 249-252 should be clarified and likewise in the discussion with exploring the CBD-sensitive subtypes of Nav channels.

Ans: Thank you for addressing the reviewer’s comments. In our investigation, the presence of CBD did not induce significant alterations in the observed INa magnitude within GH3 cells. Similarly, cannabichromene, an agonist of cannabinoid receptor 1, also failed to produce significant changes. These outcomes diverge from a prior study (Huang et al., 2021) that reported CBD’s capability to suppress the amplitude of NaV1.4 current. It thus remains unclear why this incongruity exists. One plausible explanation is that the NaV1.4 currents affected by CBD may be primarily expressed in skeletal muscle cells rather than endocrine or neuroendocrine cells. As such, it becomes imperative to undertake further investigations to elucidate whether CBD’s impact on INa is confined to specific tissue types. This paragraph related to this issue was included in the revised manuscript (indicated in lines 504-511).

- In Figure 3, no current trances were shown to support the error-bar diagram.

Ans: As advised by the reviewer, an additional panel (Figure 3A) was included in the revised version of the manuscript.

- In a conclusion remark, authors might show more potentials of their finding on exploiting CBD in basic neuroscience and clinical research.

Ans: Thanks for the comments. An additional sentence was included in the revised manuscript. That is, “Our findings are poised to offer substantial promise in terms of utilizing CBD in fundamental neuroscience and clinical research.” (indicated in lines 534-535).

Minor:

- In many places within the test, the references are not numbered, which is not adhere to the guidelines of the journal. For example, lane 236.

Ans: Goof! We did make mistakes. The references have been hence appropriately numbered in the revised manuscript.

- Also, a typo is shown all over the text that m symbol is replaced by another one.

Ans: Thanks for bringing our attention! We made mistakes. “mM” was accordingly corrected in the corresponding text of the revised manuscript.

-  In Figure 4 B and C, panels are small and disproportional to panel A.

Ans: Thanks! As suggested by the reviewer, Figures 4B and 4C were appropriately enlarged to match the size of Figure 4A.

- In Figure 6 B, the panel is disproportional and unread comparing to panel A.

Ans: As pointed out by the reviewer, Figure 6 in the revised manuscript was hence redone for clearer illustration.

Reviewer 2 Report

I have enjoyed reading this partial analysis of the effects cannabidiol on M-type and H-type currents in freshly isolated pituitary cells.  The sets of data and patterns of results that are the basis of this paper are interesting and have potential but as presented are incomplete and for that reason unconvincing.  If the authors choose to revise they will need to address and improve the areas and aspects of this manuscript that are listed below.

1.  Methods

Most of the results would be more convincing if the data sets included washout procedures and findings.  This is particularly true since only one naloxone concentration is used and this compound apparently was not employed in the I-h studies.  Furthermore, most recordings of the raw data include a surprising and unacceptable level of noise for this type of study involving small currents.  Finally, and as listed below in more detail, although it is of interest and importance to study the effects of cannabidiol on steady-state currents, there is no indication that this important requirement has been achieved in either the studies of I-M or I-h.  

2.  Data sets concerning I-h

These experiments need to be redone and supplemented.  The reasons include: a) none of the results can accurately be described as being based on steady-state current changes and this condition is important for the analysis.  b)  The I-V curve shown in Figure 7B shows marked rectification and this would not be expected for this current under the stated conditions.  c)  The oblique and awkward description of 'hysteresis' of this current is dated and should be replaced with reference to voltage- and drug-induced mode shifts with appropriate References to the papers of Siegelbaum and of Larsson et al.  

3.  Sodium current results

The finding that in your experiments cannabidiol does not alter sodium current is of definite interest but your findings need to be illustrated more clearly and convincingly in Figure 6A and in this and most other Figures your primary results need to be expressed as current densities and not simply pA.

4.  Other points

The title of your paper is misleading in that you provide very little data concerning the efficacy of cannabidiol.  Second, and more importantly, your Discussion provide absolutely no information concerning: a) how it is that you observe cannabidiol effects implying micromolar affinity and yet suggest that a drug receptor interaction is unlikely.  b) You observe an apparent cannabidiol-induced depolarization of the intrinsic voltage dependence for I-M, but at the same concentration levels of cannabidiol observe a hyperpolarization of a parameter that may relate to the voltage-dependence for activation of I-h.

This manuscript is carefully and logically organized and presented and quite well written.  However, if the authors choose to revise they could clarify importance sentences that can be found from lines 66 to 69, 116-117, 125-126, 172-173; and the entire Discussion section.

Author Response

Ans: Thanks for the insightful comments provided by the reviewer.  We will respond to the reviewer’s queries and comments in order.

  1. Methods

Most of the results would be more convincing if the data sets included washout procedures and findings.  This is particularly true since only one naloxone concentration is used and this compound apparently was not employed in the I-h studies.  Furthermore, most recordings of the raw data include a surprising and unacceptable level of noise for this type of study involving small currents.  Finally, and as listed below in more detail, although it is of interest and importance to study the effects of cannabidiol on steady-state currents, there is no indication that this important requirement has been achieved in either the studies of I-M or I-h. 

Ans:

We also performed an additional set of experiments about the effect of naloxone (30 mM). The results were included in the revised manuscript (indicated in lines 236-237). That is “Naloxone at a concentration of 30 mM) also did not cause any effect on CBD-induced decrease of IK(M).”. Moreover, Figure 3 in the revised manuscript was accordingly redone.

In terms of naloxone effect on CBD-mediated reduction of Ih, an additional series of experiments was performed. The results were hence included in the revised manuscript (indicated in lines 297-299). That is, “Moreover, in continued exposure to 3 mM CBD, further addition of naloxone (10 mM) failed to reverse CBD-mediated decrease of I­h amplitude.”.

Because some of the existing currents appeared to be small, the noise was relatively larger. However, based on the biophysical and pharmacological characteristics of various ionic currents, it is still possible to clearly observe the modulation of various ionic currents by various drugs or compounds.

Moreover, we did not clearly show significant effect of CBD on steady-state currents observed in pituitary GH3 cells, as indicated in Figure 7A and lines 294-300 of the revised manuscript.

  1. Data sets concerning I-h

These experiments need to be redone and supplemented.  The reasons include: a) none of the results can accurately be described as being based on steady-state current changes and this condition is important for the analysis.  b)  The I-V curve shown in Figure 7B shows marked rectification and this would not be expected for this current under the stated conditions.  c)  The oblique and awkward description of 'hysteresis' of this current is dated and should be replaced with reference to voltage- and drug-induced mode shifts with appropriate References to the papers of Siegelbaum and of Larsson et al. 

Ans: Thanks for the valuable comments provided by the reviewer.

What we need to emphasize is that, although the voltage-dependent hysteresis induced by Ih appear somewhat awkward, the electrical behavior itself is indeed very distinctive and significant. Its intensity contributes significantly to various forms of bursting patterns in the action potentials of many different types of excitable cells (Männikkö et al., 2005; Xiao et al., 2010).

As pointed out by the reviewer, an additional panel showing effects of CBD and CBD plus naloxone on the Ih amplitude was hence included in the revised manuscript (Figure 7A). Of note, no significant steady-state current was found during cell exposure to 3 mM CBD (indicated in lines 294-300 in the revised manuscript).

Moreover, as advised by the reviewer, an additional sentence was included in the Discussion section of the revised manuscript (lines 520-521). Two references were hence incorporated into the revised manuscript (i.e., [40,41]) (indicated in lines 651-654).

  1. Sodium current results

The finding that in your experiments cannabidiol does not alter sodium current is of definite interest but your findings need to be illustrated more clearly and convincingly in Figure 6A and in this and most other Figures your primary results need to be expressed as current densities and not simply pA.

Ans: Thanks for the comments pointed out by the reviewer. What we need to clarify is that the present study primarily focuses on the effects of drugs or compounds (e.g., cannabidiol) on individual cells before and/or after the administration. Hence, the capacitance of each individual cell itself will not have a significant impact. As a results, the original current traces in this manuscript will not be affected by the capacitance of individual cells. Thus, we believe that the amplitude of the current involved can be expressed in picoamperes (pA). However, we did include the capacitance of our study’s cells as 34 ± 6 pA (n = 32) in the text (indicated in line 132).

  1. Other points

The title of your paper is misleading in that you provide very little data concerning the efficacy of cannabidiol.  Second, and more importantly, your Discussion provide absolutely no information concerning: a) how it is that you observe cannabidiol effects implying micromolar affinity and yet suggest that a drug receptor interaction is unlikely.  b) You observe an apparent cannabidiol-induced depolarization of the intrinsic voltage dependence for I-M, but at the same concentration levels of cannabidiol observe a hyperpolarization of a parameter that may relate to the voltage-dependence for activation of I-h.

Ans: Thanks for the reviewer’s suggestion.

The title was hence appropriately changed to “Modulating M-type K+ and hyperpolarization-activated cation currents: evaluating the efficacy of cannabidiol” (indicated in lines 2-3).

In our study, during continued exposure to CBD, the decreased amplitude of IK(M) or Ih could not be reversed effectively by the subsequent addition of naloxone, a blocker of opioid receptors. Therefore, it appears unlikely that CBD-mediated modifications on these ionic currents are due to its binding to different types of opioid receptors.

         In our study, we did observe the inhibitory effects of CBD on IK(M) and IK(erg) in pituitary GH3 cells. Hence, the CBD’s perturbations on different types of ionic currents appear to be complex.

Reviewer 3 Report

The study performed by Yen-Chin Liu and coauthors demonstrates the effects of cannabidiol on the potassium and sodium currents in pituitary GH3 cells. Cannabidiol and its derivatives are considered promising drugs for the treatment of numerous psychiatric and neurological diseases. Undoubtedly, the investigation of cannabidiol-mediated effects on excitable cells is an actual task of pharmacology and medicine. Therefore, the obtained results will be of interest to a wide range of readers. I have only some comments regarding the submitted manuscript. 

- In its current form, the article seems like a case report demonstrating only the effects of the studied drugs. Some molecular mechanisms of cannabinoids have been reported or hypothesized in the literature. Moreover, the authors write in Introduction "investigate the possible underlying mechanisms". However, the manuscript lacks any mechanisms. I would recommend adding the possible targets of cannabidiol, and their downstream signaling pathways, in terms of the influence on ion channels. This information should be added to Introduction and Discussion sections. 

- Please carefully check and correct concentrations of the used drugs throughout the manuscript;

- Information about the drug application, especially about incubation time, is required.

- Please indicate in each figure legend which statistical test was used. 

Only minor editing can be recommended. 

Author Response

Ans: Thanks for the insightful comments provided by the reviewer. 

Although some molecular mechanisms of cannabinoids have been demonstrated, we presented the novel findings showing that the presence of cannabinoid exerted an inhibitory action on the amplitude and gating of both IK(M) and Ih.  We do not believe that our paper is merely a case report.  Our findings will be poised to offer substantial promise in terms of utilizing cannabinoids in fundamental neuroscience and clinical research.

         Moreover, our findings suggest that some important targets of cannabinoids are membrane ion channels.  As for how cannabinoids modulate these ionic currents and potentially disrupt specific downstream signaling pathways, further in-depth research is warranted in the future.  The text was hence included in the revised manuscript (indicated in lines 520-521).

         To address the concerns of the reviewer, we have appropriately refined the term “… underlying mechanism” (indicated in lines 75-77 of the revise manuscript).  That is, “Therefore, based on the aforementioned information, our aim was to investigate the effects of CBD or other related compounds on perturbations of various ionic currents (such as INa, IK(M), Ih, and IK(erg)) in pituitary GH3 cells.”

- Please carefully check and correct concentrations of the used drugs throughout the manuscript;

Ans: Thanks for the reviewer’s comments. The concentration of the used drugs has been corrected appropriately in the revised manuscript.

- Information about the drug application, especially about incubation time, is required.

Ans: As advised by the reviewer, the statement regarding the time of drug or compound administration during each experiment was included in the text of the corresponding results.

- Please indicate in each figure legend which statistical test was used.

Ans: Following the reviewer’s suggestion, we have incorporated the details of the statistical method into the legend of the corresponding figure in the revised manuscript.

Round 2

Reviewer 2 Report

Thank you for considering my original comments.  I note and appreciate the fact that you have altered some Figures, added some References and done further editing of the original manuscript.  However, and unfortunately you have not appreciated or chosen not to make changes that deal correctly or sufficiently with some of my major comments.  These include:

1.  I pointed out that from the description of your results and from the raw data that was included you have no basis for making any meaningful statements about shifts in the voltage dependence of steady state activation curves, or the slopes of those relationships.  These statements are made twice in the Abstract and in other places in your R1 manuscript.  The information that can be obtained from a true steady-state activation curve does, indeed, provide a basis for making conclusions regarding the effects of drugs or experimental molecules on the gaiting process of the ion channel in question.  None of your raw data illustrate the absolute requirement of the onset of the current change having reached steady-state.  Accordingly none of your conclusions can be accepted at face value and some may be inaccurate.

2.  In my original review I also pointed out that the signal-to-noise ratio of the current records in some of your key experimental data sets was unusually high and should be improved.  I fully appreciate that the current changes that you have measured are small but this is perhaps the main reason to work meaningfully and effectively using both accepted grounding techniques, electrode shielding and post processing to present your data more clearly and convincingly.  

3.  In relation to your sodium current measurements and in particular based on the interesting and possibly important negative findings, I asked for these and other data sets to be presented as current densities.  This is a standard and conventional format for presenting data concerning transient currents and not a minor request.

4.  In the R1 manuscript many of the drug or experimental compounds are described as being applied in milli molar concentrations.  In most cases I presume you mean micro molar and this level of attention to detail is required for appropriate and positive review of an R1 manuscript.

The authors have improved their choice and use of Scientific English.

Author Response

Thank you for considering my original comments.  I note and appreciate the fact that you have altered some Figures, added some References and done further editing of the original manuscript.  However, and unfortunately you have not appreciated or chosen not to make changes that deal correctly or sufficiently with some of my major comments.  These include:

Ans:  Thanks for the insightful and valuable comments pointed out by the reviewer.

  1. I pointed out that from the description of your results and from the raw data that was included you have no basis for making any meaningful statements about shifts in the voltage dependence of steady state activation curves, or the slopes of those relationships. These statements are made twice in the Abstract and in other places in your R1 manuscript.  The information that can be obtained from a true steady-state activation curve does, indeed, provide a basis for making conclusions regarding the effects of drugs or experimental molecules on the gaiting process of the ion channel in question.  None of your raw data illustrate the absolute requirement of the onset of the current change having reached steady-state.  Accordingly none of your conclusions can be accepted at face value and some may be inaccurate.

Ans:  Thanks for the insightful comments.  An additional description was hence included in the text of the revised manuscript.  That is, “However, we found no clear adjustments in the slope factor (k value or steepness) of the curve when exposed to CBD, suggesting that there was no change in the gating charge of the curve during its exposure. Alternatively, the CBD-induced inhibition of IK(M) in unclamped cells could depend on the pre-existing resting membrane.” (as indicated in lines 221-225 of the revised manuscript). 

Similarly, an additional statement was included in the revised manuscript.  That is, “Therefore, it appears likely that no change in the gating charge of Ih activation curve was altered by CBD exposure.  The responsiveness of Ih to CBD is influenced by varying levels of pre-existing resting membrane potential.” (as indicated in lines 323-326 of the revised manuscript).

  1. In my original review I also pointed out that the signal-to-noise ratio of the current records in some of your key experimental data sets was unusually high and should be improved. I fully appreciate that the current changes that you have measured are small but this is perhaps the main reason to work meaningfully and effectively using both accepted grounding techniques, electrode shielding and post processing to present your data more clearly and convincingly. 

Ans:  Thanks for the insightful comments regarding our experiments.  Indeed, both accepted grounding techniques and electrode shielding are crucial for performing electrophysiological measurements. We will do our best to enhance the signal-to-noise ratio of the experimentally observed current records.

  1. In relation to your sodium current measurements and in particular based on the interesting and possibly important negative findings, I asked for these and other data sets to be presented as current densities. This is a standard and conventional format for presenting data concerning transient currents and not a minor request.

Ans:  Thanks for the comments provided by the reviewer. Of note, the current traces in the manuscript were changed to current densities as advised.  The majority of figures in this version of the revised manuscript have hence extensively been redone with the replacement of pA with pA/pF in the labeling of y-axis.

  1. In the R1 manuscript many of the drug or experimental compounds are described as being applied in milli molar concentrations. In most cases I presume you mean micro molar and this level of attention to detail is required for appropriate and positive review of an R1 manuscript.

Ans:  Thanks for bringing our attention. In this version of the revised manuscript, “mM” has been appropriately changed to “mM”. (e.g., line 290 and 321 in the revised manuscript)

Reviewer 3 Report

My comments have been addressed. Some misleading issues, including the aim of the research and title, have been corrected. Regarding the title, I would recommend rewriting it (for instance "Cannabidiol modulates... " or similarly). 

Only minor corrections are still required. 

Author Response

My comments have been addressed. Some misleading issues, including the aim of the research and title, have been corrected. Regarding the title, I would recommend rewriting it (for instance "Cannabidiol modulates... " or similarly).

Ans: Thanks for the comment provided by the reviewer. Hence, the title was appropriately changed to “Cannabidiol modulates M-type K+ and hyperpolarization-activated cation currents” (indicated in lines 2-3 of the revised manuscript).

Round 3

Reviewer 2 Report

Thank you for considering my comments on your R1 manuscript and making additional changes in the text and revising some of the Figures.  I appreciate this and consider the R1 manuscript to be improved.  However, there is an important concept and principle that the R1 manuscript and your responses to me still do not present clearly convincingly.  This is based on the details for constructing a steady-state activation curve from experimental data, and the  information that can be obtained from this important data.  In both of my reviews, I have pointed out that your raw experimental current records clearly show that the voltage clamp pulses have not been long enough to drive the ion transfer mechanism to steady-state.  Accordingly, the activation curve that you obtain and present do not contain reliable information concerning the exact voltage dependence for activation of this current or the steepness or sensitivity of the gating mechanisms as indicate by the slope factor.  I make and insist on these points because as we agree only very small changes in net current (modulated by M-type potassium channels) are known to be functionally important both physiologically and in pathophysiological settings in small cells such as the ones that you have studied, pituitary GH3 cells.  Please consider rewording your manuscript and including this in your Limitations section.

Author Response

Ans:  We appreciate the insightful and valuable comments pointed out by the reviewers. 

We agree with the reviewer’s perspective.  Indeed, even a slight variation in M-type potassium currents can have a profound impact on the physiological or even pathological function of small cells.  Moreover, concerning the steady-state activation curve of the M-type potassium current, it is crucial to have a sufficiently long step command duration to accurately assess the lope factor and V1/2 values.  This is essential for a more precise determination of the steady-state activation curve of this current, followed by the application of the Boltzmann equation to deduce V1/2 and the slope factor, subsequently allowing us to predict the gating charge value.

         However, in these cells we have been investigating (such as GH3 cells), when we employed a voltage-clamp protocol that exceeds 1 second, we often observed the appearance of the inactivation process of other delayed rectifier potassium currents.  The primary reason behind this is that GH3 cells, in addition to M-type potassium currents, still possess various types of delayed rectifier potassium currents elicited by membrane depolarization.  This is a common issue encountered when conducting experiments on native excitable cells.  Using cells with KCNQx-encoded may help to mitigate these complications.

         Therefore, to address the reviewer’s comments, we have made appropriate modifications to the steady-state activation curve in our manuscript.  Specifically, we have replaced the ‘stead-state activation curve of the current” with the “quasi-steady-state activation curve of the current.”  The prefix “quasi-“ is usually in English language to indicate that something resembles or is similar to something else but is not exactly the same.  It is often used to convey a sense of partial similarity or resemblance.

Additionally, we have organized the explanation mentioned above and then incorporated it into the Discussion section of this new version of the revised manuscript to draw the readers’ attention to it (as indicated in lines 501-515).  That is, “It needs to be mentioned that concerning the steady-state activation curve of IK(M), it is important to have a sufficiently long step comment duration to accurately assess the slope factor and V1/2 values.  This is essential for a more precise determination of the activation curve of this current, followed by the application of the Boltzmann equation to deduce V1/2 and the slope factor, subsequently allowing use to predict the gating charge value.  However, in the cells we have investigated (such as GH3 cells), when we employed a voltage-clamp protocol that exceeds 1 sec, we often observed the appearance of the inactivation process from other types of K+ currents.  This phenomenon, to certain extent, led to confusion with the inactivation of other delayed-rectifier K+ currents.  The primary reason behind this is that GH3 cells, in addition to IK(M), still possess various types of delayed-rectifier K+ currents.  This is a common issue encountered when conducting experiments on native excitable cells.  Using cells with KCNQx-encoded currents may help mitigate these complications.  Nonetheless, the exposure of CBD to GH3 cells did suppress the strength of IK(M) and shift the quasi-steady-state activation curve of IK(M) to a more depolarized potential without affecting the curve’s slope factor.”

Therefore, it is important to note that the presence of cannabidiol does indeed suppress the strength of the M-type potassium current and shifts the quasi-steady-state activation curve of IK(M) to a more depolarized potential without affecting the curve’s slope factor.